## PERSPECTIVE

# Electroreceptors with a different kind of buzz

Geng-Lin Li[1] and Henrique von Gersdorff[2]

[1] *Department of Otorhinolaryngology, Eye and ENT Hospital, Fudan University, Shanghai, China*

[2] *Vollum Institute, Oregon Health & Science University, Portland, OR, USA*

Email: genglin.li@fdeent.org; vongersd@ohsu.edu

Handling Editors: Katalin Toth & Samuel Young

The peer review history is available in the Supporting Information section of this article (https://doi.org/10.1113/JP289354#support-information-section).

Electric fish in Africa and South America have evolved electroreceptors and electric organs that permit them to detect and produce electric fields. They use these electric fields to communicate, navigate and find prey even as nocturnal animals inhabiting muddy rivers where visual cues are limited. The emittance of these dipole electric fields is sophisticated in nature, with different fish species capable of generating electric organ discharges (EODs) with diverse waveforms at frequencies up to 20 kHz. To detect electric fields, electric fish evolved specialized electroreceptors along their body. Given that dipole electric fields decay sharply with distance ($d$) as $1/d^3$, some of these electroreceptors must be highly sensitive, thus requiring them to be well tuned to pick up small signals from noise, as seen in other sensory modalities. However, how these electroreceptors can be tuned to detect weak electric signals oscillating at up to 20 kHz is poorly understood. This is particularly puzzling given that electrical tuning in non-mammalian auditory hair cells is limited to less than 5 kHz (Fettiplace, 2020).

Among different types of electroreceptors, Knollenorgans are one type of tuberous electroreceptors that are tuned to higher frequencies (Bell, 1990; Fig. 1*A*). In an article in this issue of *The Journal of Physiology*, Raman and Hopkins (2025) stimulated Knollenorgans in mormyrid fish with sinusoidal electric fields of different frequencies and amplitudes, recording *in vivo* spike-like responses extracellularly. This allowed them to build tuning curves for individual Knollenorgans, in a fashion similar to constructing tuning curves for individual auditory afferent fibres with pure tone stimulation. Although these tuning curves are not very sharp with a typical quality factor $Q_{BP} < 1$ (Frolov & Li, 2017), their best frequencies are found to be as high as 16 kHz. Raman and Hopkins (2025) find that different fish species have different tuning curves matched to the spectrum of EODs that the fish emit, a strong indication that this electrical tuning has evolved to mediate conspecific communication. Moreover, arbitrary electrical noise stimuli applied to Knollenorgan electroreceptors evoked precisely timed spike patterns, which were well replicated by a model of linear filtering, rectification and spike thresholding. These electrically evoked spikes in Knollenorgans are reliable and precise on a time scale of less than 0.1 ms and can thus transmit signals above 10 kHz.

Of course, the burning mechanistic question is how this electrical tuning arises given our current understanding of excitable membranes in analogous hair cells. Electrical tuning in non-mammalian auditory hair cells stems from $Ca^{2+}$ currents closely coupled with $Ca^{2+}$-dependent $K^+$ currents. This produces membrane voltage oscillations that resonate at a particular frequency (Fig. 1*B*, grey), which matches the best (or characteristic) frequency in tuning curves. Results from avian, amphibian and reptilian species have suggested that resonant frequencies in hair cells cannot exceed 5 kHz, even in warm-blooded avian species, because this would require not only a very large $Ca^{2+}$ current to drive steep voltage changes but also a prohibitive amount of ATP to subsequently extrude the $Ca^{2+}$ ions. Moreover, the membrane time constant ($\tau = R_m C_m$, where $R_m$ is membrane resistance and $C_m$ is membrane capacitance) would have to be in the submillisecond range, which presumably requires an extremely low $C_m$ and $R_m$. However, Knollenorgan electroreceptors are densely studded with microvilli (Fig. 1*A*), which will produce a large surface area and thus a large $C_m$.

Remarkably, Knollenorgans fire spikes whose waveforms are diverse among different species, as discussed by Raman and Hopkins (2025). Notably, these spikes of Knollenorgans are not blocked by tetrodotoxin, so they are presumably $Ca^{2+}$ action potentials. In Fig. 1*B*, we propose a putative model using a combination of electrical resonance and spike generation (depicted in grey and black): strongly attenuated subthreshold membrane potential oscillations would resonate at a particular stimulus frequency, and when the threshold is crossed, a single $Ca^{2+}$ action potential is fired (Rutherford & Roberts, 2009). Thus, a small signal that matches the resonant frequency of a particular electroreceptor would produce a single spike. The spike would be ultranarrow, due perhaps to a large $Ca^{2+}$ current with unusually fast activation and inactivation kinetics, followed by large $K^+$ currents during the spike downstroke. The height of the depolarizing plateau during the stimulus is $V_{plateau} = R_m I_{Transduction}$, where $I_{Transduction}$ is the stimulus-induced current. If $R_m$ is very small, because $\tau$ needs to be small, a large $I_{Transduction}$ is presumably required for $V_{plateau}$ to be near the threshold. Alternatively, $V_{Threshold}$ may be strategically positioned closely above $V_{Rest}$, although it needs to be safely placed to avoid false-positive spikes generated by random membrane potential fluctuations. A relatively large $I_{Transduction}$ perhaps produced by even a very weak stimulus, may be mediated by a large voltage-gated $Ca^{2+}$ current in the electroreceptor, which expresses multiple mitochondria just underneath the microvilli. One can also speculate that the volume outside of the cell may have a high external free $Ca^{2+}$ ion concentration and a high pH (Cho & von Gersdorff, 2014), so the $Ca^{2+}$ current may indeed be relatively large even for small deviations from $V_{Rest}$.

The elegant new work of Raman and Hopkins (2025) reminds us how little we know about the cellular biophysics of the Knollenorgans. It thus challenges us to push the current conceptual boundaries, develop more realistic conductance-based models of these specialized cells, and test them with further inquiries through direct intracellular recording. If subthreshold membrane oscillations exist in Knollenorgan electroreceptors, it would be remarkable if they indeed occur in the range of 5–10 kHz. This does require a bit of 'suspension of disbelief', but we should keep an open mind since electrical sensing may provide an immense survival

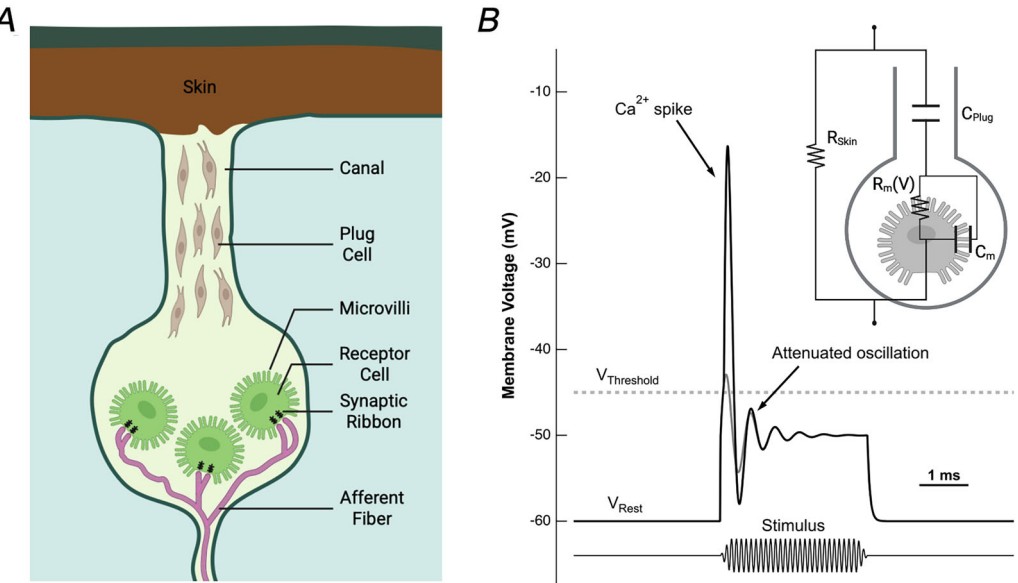

**Figure 1. Anatomy and function of tuberous electroreceptor cells in the Knollenorgan**
*A*, schematic drawing of the Knollenorgan. The electroreceptor cells (green) are isolated from the external environment by a layer of skin and plug cells in the canal. Electroreceptor cells are studded with microvilli, and on their basal pole, a single afferent fibre branches out and makes ribbon-type synapses onto multiple electroreceptor cells. The high sensitivity of this single afferent fibre to small amplitude electric fields may originate in part from these multiple ribbon synapses, because coincident small synaptic events evoked by weak electric fields may summate to trigger a spike, whereas random small events from spontaneous synaptic release or noise will remain subthreshold. *B*, signalling of the electroreceptor in the Knollenorgan. In response to an electrical field stimulus (e.g. electric organ discharge, EOD), the electroreceptor depolarizes and may exhibit attenuated oscillations due to its electrical tuning (in grey). When the $V_{Threshold}$ is crossed, a single $Ca^{2+}$ spike is fired, overlaid on top of the attenuated oscillation (in black). The inset depicts a simplified electrical circuit from *A* with two RC filters ($R_{Skin}$ and $C_{Plug}$, $R_m(V)$ and $C_m$) that together produce a bandpass filter to replicate the overall V-shaped electrical tuning curves observed in the Knollenorgan (Lyons-Warren et al., 2012). Here $R_m(V)$ represents a voltage-dependent resistor that gives rise to an electric resonance based on voltage-gated $Ca^{2+}$ currents and $Ca^{2+}$-dependent $K^+$ currents.

advantage for electric fish. It afforded them extraordinary diversification in species – a beautiful example of convergent evolution from a non-electrogenic common ancestor some 20 million years ago, well after Pangea began splitting Africa from South America more than 200 million years ago.

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

## Additional information

### Competing interests

The authors declare they have no competing interests.

### Author contributions

Both authors have read and approved the final version of this manuscript and agree to be accountable for all aspects of the work in ensuring that questions related to the accuracy or integrity of any part of the work are appropriately investigated and resolved. All persons designated as authors qualify for authorship, and all those who qualify for authorship are listed.

### Funding

No funding was received for this work.

## Keywords

bandpass filter, electric fish, electric resonance, electric tuning, electroreceptors, Knollenorgans, species diversity

## Supporting information

Additional supporting information can be found online in the Supporting Information section at the end of the HTML view of the article. Supporting information files available:

**Peer Review History**

