## [Peer Review History · The Journal of Physiology]

Electroreceptors with a different kind of buzz

Henrique von Gersdorff and Geng-Lin Li

DOI: 10.1113/JP289354

Corresponding author(s): Geng-Lin Li (genglin.li@fdeent.org)

Review Timeline:

Submission Date:	02-Jul-2025
Editorial Decision:	17-Jul-2025
Revision Received:	05-Sep-2025
Editorial Decision:	24-Sep-2025
Revision Received:	14-Oct-2025
Accepted:	27-Oct-2025

Senior Editor: Katalin Toth

Reviewing Editor: Samuel Young

Transaction Report:

Dear Dr Li,

Re: JP-P-2025-289354 "Electroreceptors with a different kind of buzz" by Henrique von Gersdorff and Geng-Lin Li

Thank you for submitting your manuscript to The Journal of Physiology. It has been assessed by a Reviewing Editor and by 1 expert referee and we are pleased to tell you that it is potentially acceptable for publication following satisfactory major revision.

The review comments are copied at the end of this email.

Please address all the points raised and incorporate all requested revisions or explain in your Response to Referees why a change has not been made. We hope you will find the comments helpful and that you will be able to return your revised manuscript within 2 weeks. If you require longer than this, please contact journal staff: jp@physoc.org. Please note that this letter does not constitute a guarantee for acceptance of your revised manuscript.

REVISION CHECKLIST:

IMPORTANT POINTS TO NOTE WHEN REVISING YOUR MANUSCRIPT:

LANGUAGE EDITING AND SUPPORT FOR PUBLICATION: If you would like help with English language editing, or other article preparation support, Wiley Editing Services offers expert help, including English Language Editing, as well as translation, manuscript formatting, and figure formatting at www.wileyauthors.com/eo/preparation. You can also find resources for Preparing Your Article for general guidance about writing and preparing your manuscript at www.wileyauthors.com/eo/prepresources.

We look forward to receiving your revised submission.

Yours sincerely,

Katalin Toth
Senior Editor
The Journal of Physiology

EDITOR COMMENTS

The reviewers have significant concerns with the perspective, with the major concern that the perspective does not explain or summarize the paper. Based on the reviewers positive and constructive comments a major text revision that accurately explains and summarize the manuscript is needed.

REFeree COMMENTS

Referee #1:

Thank you for the opportunity to comment on this Perspective. Both authors (Raman and Hopkins) have collaborated on this response. Our general comment is that the Perspective does not explain or briefly summarize the content of the paper. Instead, it makes reference to behaviors that are not accomplished by the tuberous electroreceptors studied here (predator evasion, sound localization) and skips to mechanism, which was not investigated here (even though the Perspective makes it seem like that was the focus of the present work). It seems somewhat odd that most of the actual discoveries of the paper are either not mentioned, such as the correlation between electroreceptor tuning and EOD power spectrum, or are confused, such as the form of the filter. It might be helpful to provide a brief and accurate summary of what was actually done before pursuing the question of mechanism that interests the authors.

Below are several points identifying erroneous statements that should be corrected.

Specific points.

1. Line 12, "By emitting electric fields that oscillate at high frequencies...." In the (mormyrid) fish studied here, electric fields don't oscillate, but are pulses. It would therefore be more accurate to state that electric fields from EODs (which are pulses) have high spectral frequencies.

2. Lines 12-15, "some species are thought to employ a type of 'signal cloaking'... (Stoddard and Markram)" The paper by Stoddard and Markham (note spelling: not Markram) refers not to the "production" of signals of a given frequency but to the "avoidance" of signals with power at low frequencies (i.e. any signal in the 0 to 100 or typically 0 to 30 Hz), which is the range of sensitivity of electroreceptive predators like catfishes (Clariids) that are known to prey on electric fish. The present study is about tuberous electroreceptors and high spectral frequency EODs, whose evolution is not driven by signal cloaking but is more likely to have been driven by the needs for communication and electrolocation. Thus, bringing up predator avoidance in the opening paragraph does not accurately introduce the work to be discussed.

3. Line 16, "Tuberous electroreceptors tuned to high frequencies are located in Knollenorgans." Knollenorgans "are" a type of tuberous electroreceptors, i.e., one is not inside the other. Perhaps what is meant is that Knollenorgans are tuberous electroreceptors that contain sensory cells. Knollenorgans themselves are located in the skin of the head and body.

4. Lines 16-19, "Knollenorgans... are also hyper-sensitive to very low intensity electric fields... important because the fish are basically electric dipoles and static dipole fields decay sharply with distance (d) as $1/d^3$ " The mention of inverse cube rate of electric field decline with distance does not justify the sensitivity of Knollenorgans, since mormyromast receptors sensing the fish's own EOD are much less sensitive. It would be more correct to say that the sensitivity of Knollenorgans is needed for detection of communication signals at a distance.

5. Lines 19-21: "This high sensitivity may originate in part from the multiple ribbon-type synapses that electroreceptors make with single afferent fibers that project to the brain (Fig. 1A)." This assertion does not make sense. Ribbon synapses (an output) seem unlikely to generate a high sensitivity to electric fields (an input).

6. Line 21-23. "Knollenorgan electroreceptors have unusual properties: a single sub-millisecond spike..." This description of "the" spike from Lyons-Warren et al. 2012, who studied a different species (*B. brachyistius*), which is tuned to lower frequencies than the ones in the Raman-Hopkins paper, and which has a correspondingly different spike shape (slower, without ringing). A major point of the present paper is that attributes of Knollenorgan responses are species specific, like the EOD. Therefore, instead of taking the spike from the Lyons-Warren paper as a prototype, it would be correct (and even necessary) to emphasize the species diversity and indicate that the spike shapes vary across species, the same way that a hippocampal spike is not representative of a Purkinje cell spike. As described in the discussion of the published paper, many of the spike-like receptor potentials in the present study actually had larger oscillations. An example from the genus *Campylomormyrus* is present in Figure 5 of a publication from the Hopkins lab in the Cornell Synapse. We include the link here just so that the authors of the Perspective can get a sense of the diversity we are talking about. https://www.researchgate.net/profile/Carl-Hopkins-2/publication/265926422_Electric_Signaling_and_Electroreception_Properties_in_Electric_Fishes_of_the_Genus_Campylomormyrus_Mormyridae_ARTICLE/links/54d0e000cf20323c21a00c3/Electric-Signaling-and-Electroreception-Properties-in-Electric-Fishes-of-the-Genus-Campylomormyrus-Mormyridae-ARTICLE.pdf

7. Lines 23-25: "The microsecond timing difference of these spikes occurring in different electroreceptors along the fish's body allow the brain to calculate the location and speed of nearby objects." There are some fundamental misunderstandings here. First, the EOD does not propagate as a wave like sound does, but it is an electrostatic signal, moving essentially at the speed of light. Second, there is no evidence for timing being used by these electroreceptors to measure speed of objects. The idea that appears to be missing here (and possibly the idea that the authors may be going for?) is that high frequency tuning is matched to the power spectrum of the EOD, and that the Knollenorgan sensory pathway involves specializations (electrical synapses, large diameter fibers, synapses on cell bodies etc.) so that there is very little jitter in the temporal coding of the EOD. This lack of jitter permits the EOD to be recognized by the timing cues. But that is different from the "microsecond timing difference" of sound localization, which seems to be referenced here.

8. Line 31 and line 67 "Raman and Carlson." The paper is by Raman and Hopkins, not Raman and Carlson.

9. Lines 33-34. "The authors hypothesize that the mechanisms are linear bandpass filtering coupled with non-linear thresholding...." This statement is incorrect.

a. Contrary to the statement made by the authors, Figure 1 of the paper shows that a simple bandpass filter is "not" sufficient to describe the tuning curves. It shows that there is instead some process that mimics an electrical resonance. This should be corrected.

b. The circuit in the perspective's Figure 1B is the diagram proposed (by Carlson) in Lyons-Warren et al. (2012), for *B. brachyistius*, a species not studied in the Raman-Hopkins paper. This circuit is "not" able to describe the results in the present paper. Therefore, the statement in lines 33-34 should be modified or corrected, and the circuit diagram probably should be changed or removed.

10. Lines 35-37. "they propose a subthreshold electrical resonance in Knollenorgan electroreceptors that stems from Ca^{2+} currents closely coupled with Ca^{2+} -dependent K^{+} currents." This statement is inaccurate. No model of specific currents is proposed. It should be made clear in the text that the present paper makes no claims about mechanism because at present no mechanistic data are available. The paper (by intention) presents the phenomenology that directs the hunt for mechanism. The hypotheses tested in the paper were (1) that the high frequencies of the EODs of these three species would be predictive of correspondingly high-frequency tuned Knollenorgans (supported), and (2) that the transduction process might be approximated as a linear filter (supported with the revision of including rectification and the non-linear spike generating mechanism). These hypotheses should be distinguished from the subsequent questions that became possible to articulate once the paper's results were in hand, namely, how (mechanistically) can electroreceptors achieve tuning up to such high frequencies? The data presented in the paper offer a constraint, namely, whatever the mechanism actually is, its output must produce something that mimics a bandpass filter scaled by an electrical resonance. The relevant section of the Discussion begins with the statement "The puzzle that persists is mechanistic," followed by a literature review of pertinent information from other systems. Ca and $K(Ca)$ currents are discussed in that context, but no claims or proposals are made in the absence of data.

11. Lines 49-51: "Figure 1B shows a putative intracellular recording from a single electroreceptor." This is a far overstated claim. This is not a putative intracellular recording, but is the output of a model proposed for a different species, and the only data available (i.e., the Raman-Hopkins paper) indicate a very different mechanism. Along the lines of the previous point, an acknowledgment of species diversity is crucial in the context of offering perspective on a paper that is about species diversity.

12. Figure 1A: it is not clear where that schematic came from. A reference is needed. It may be worth noting that the sensory cells in the EMs that from Szabo have microvilli all around, not just on the apical end. (In this regard they are unlike mechanosensitive hair cells).

13. Minor. Line 9-10. "murky and dark rivers of Africa." The authors might consider "murky and dim rivers."

END OF COMMENTS

Dear Dr. Toth,

Many thanks for sending us the reviewer comments (in blue), and in below is our point-by-point response (in black). We have extensively edited our text and followed almost all the suggestions of the reviewers. We thank the reviewers for their thoughtful, detailed and helpful suggestions.

EDITOR COMMENTS

The reviewers have significant concerns with the perspective, with the major concern that the perspective does not explain or summarize the paper. Based on the reviewers positive and constructive comments a major text revision that accurately explains and summarize the manuscript is needed.

After an introductory paragraph we have extensively rewritten the second paragraph to summarize and highlight all the major findings of Raman and Hopkins:

“Among different types of electroreceptors, Knollenorgans are one type of tuberous electroreceptors that are tuned to higher frequencies (Bell,1990, Fig. 1A). In this issue, Raman and Hopkins (2025) stimulated Knollenorgans in mormyrid fish with sinusoidal electric fields of different frequencies and amplitudes, recording *in vivo* spike-like responses extracellularly. This allowed them to build tuning curves for individual Knollenorgans, in a fashion similar to constructing tuning curves for individual auditory afferent fibers with pure tone stimulation. Although these tuning curves are not very sharp with a typical quality factor $Q_{BP} < 1$ (Frolov and Li, 2017), their best frequencies are found to be as high as 16 kHz. These tuning curves matched very well with the spectrum of EODs these fish emit, a strong indication that this electrical tuning has evolved to mediate conspecific communication. Moreover, a combination nonlinear thresholding and rectification produces well-timed spikes in Knollenorgan electroreceptors that vary in waveform across different species of weakly electric fish. These electrically evoked spikes in Knollenorgans are reliable and precise on a time scale of less than 0.1 ms and can thus transmit signals above 10 kHz.”

We also now state in the fourth paragraph: “Remarkably, Knollenorgans fire spikes whose waveforms are diverse among different species, as shown by Raman and Hopkins (2025). Notably, these spikes of Knollenorgans are not blocked by TTX, so they are presumably Ca^{2+} action potentials.”

REFEREE COMMENTS

Referee #1:

Thank you for the opportunity to comment on this Perspective. Both authors (Raman and Hopkins) have collaborated on this response. Our general comment is that the Perspective does not explain or briefly summarize the content of the paper. Instead, it makes reference to behaviors that are not accomplished by the tuberous electroreceptors studied here (predator evasion, sound localization) and skips to mechanism, which was not investigated here (even though the Perspective makes it seem like that was the focus of the present work). It seems somewhat odd that most of the actual discoveries of the paper are either not mentioned, such as the correlation between electroreceptor tuning and EOD power spectrum, or are confused, such as the form of the filter. It might be helpful to provide a brief and accurate summary of what was actually done before pursuing the question of mechanism that interests the authors.

See our response above to the Editor.

We agree with this point and have extensively rewritten the second paragraph of the piece to mention all major findings and we also removed all reference to predator evasion and sound localization. We now state that: "These tuning curves matched very well with the spectrum of EODs these fish emit, a strong indication that this electrical tuning has evolved to mediate conspecific communication."

Below are several points identifying erroneous statements that should be corrected.

Specific points.

1. Line 12, "By emitting electric fields that oscillate at high frequencies..." In the (mormyrid) fish studied here, electric fields don't oscillate, but are pulses. It would therefore be more accurate to state that electric fields from EODs (which are pulses) have high spectral frequencies.

We removed this sentence and changed extensively the text to now read: "The emittance of these dipole electric fields is sophisticated in nature, with different fish species capable of generating electric organ discharges (EODs) with diverse waveforms at frequencies up to 20 kHz."

2. Lines 12-15, "some species are thought to employ a type of 'signal cloaking'... (Stoddard and Markram)" The paper by Stoddard and Markham (note spelling: not Markram) refers not to the *production* of signals of a given frequency but to the *avoidance* of signals with power at low frequencies (i.e. any signal in the 0 to 100 or typically 0 to 30 Hz), which is the range of sensitivity of electroreceptive predators like catfishes (Clariids) that are known to prey on electric fish. The present study is about tuberous electroreceptors and high spectral frequency EODs, whose evolution is not driven by signal cloaking but is more likely to have been driven by the needs for communication and electrolocation. Thus, bringing up predator avoidance in the opening paragraph does not accurately introduce the work to be discussed.

We agree and thus we removed the Stoddard and Markham paper and all reference to signal cloaking. In the second paragraph we now state: "These tuning curves matched very well with the spectrum of EODs these fish emit, a strong indication that this electrical tuning has evolved to mediate conspecific communication."

3. Line 16, "Tuberous electroreceptors tuned to high frequencies are located in Knollenorgans." Knollenorgans *are* a type of tuberous electroreceptors, i.e., one is not inside the other. Perhaps what is meant is that Knollenorgans are tuberous electroreceptors that contain sensory cells. Knollenorgans themselves are located in the skin of the head and body.

Text revised as suggested:

"Among different types of electroreceptors, Knollenorgans are one type of tuberous electroreceptors that are tuned to high frequencies (Bell, 1990, Fig. 1A)."

4. Lines 16-19. "Knollenorgans... are also hyper-sensitive to very low intensity electric fields... important because the fish are basically electric dipoles and static dipole fields decay sharply with distance (d) as $1/d^3$ " The mention of inverse cube rate of electric field decline with distance does not justify the sensitivity of Knollenorgans, since mormyromast receptors sensing the fish's own EOD are much less sensitive. It would be more correct to say that the sensitivity of Knollenorgans is needed for detection of communication signals at a distance.

Text revised as suggested:

"Given that electric fields decay sharply with distance (d) as $1/d^3$, these electroreceptors are believed to be highly sensitive, especially for electrical sensing at distance, requiring them to be well tuned to pick up signals from noise, as seen in many other sensory modalities."

5. Lines 19-21: "This high sensitivity may originate in part from the multiple ribbon-type synapses that electroreceptors make with single afferent fibers that project to the brain (Fig. 1A)." This assertion does not make sense. Ribbon synapses (an output) seem unlikely to generate a high sensitivity to electric fields (an input).

We have made this point clearer and moved it to the figure caption where we say:

"The high sensitivity of this single afferent fiber to small electric fields may originate in part from these multiple ribbon synapses, because coincident small synaptic events evoked by weak electric fields may summate to trigger a spike, whereas random small events from spontaneous synaptic release or noise will remain subthreshold."

6. Line 21-23. "Knollenorgan electroreceptors have unusual properties: a single sub-millisecond spike...." This description of "the" spike from Lyons-Warren et al. 2012, who studied a different species (*B. brachyistius*), which is tuned to lower frequencies than the ones in the Raman-Hopkins paper, and which has a correspondingly different spike shape (slower, without ringing). A major point of the present paper is that attributes of Knollenorgan responses are species specific, like the EOD. Therefore, instead of taking the spike from the Lyons-Warren paper as a prototype, it would be correct (and even necessary) to emphasize the species diversity and indicate that the spike shapes vary across species, the same way that a hippocampal spike is not representative of a Purkinje cell spike. As described in the discussion of the published paper, many of the spike-like receptor potentials in the present study actually had larger oscillations. An example from the genus *Campylomormyrus* is present in Figure 5 of a publication from the Hopkins lab in the Cornell Synapse. We include the link here just so that the authors of the Perspective can get a sense of the diversity we are talking about. https://www.researchgate.net/profile/Carl-Hopkins-2/publication/265926422_Electric_Signaling_and_Electroreception_Properties_in_Electric_Fishes_of_the_Genus_Campylomormyrus_Mormyridae_ARTICLE/links/54d0e000cf20323c21a00c3/Electric-Signaling-and-Electroreception-Properties-in-Electric-Fishes-of-the-Genus-Campylomormyrus-Mormyridae-ARTICLE.pdf

Text revised as suggested.

"Remarkably, Knollenorgans fire spikes whose waveforms are diverse among different species, as shown by Raman and Hopkins (2025)".

7. Lines 23-25: "The microsecond timing difference of these spikes occurring in different electroreceptors along the fish's body allow the brain to calculate the location and speed of nearby objects." There are some fundamental misunderstandings here. First, the EOD does not propagate as a wave like sound does, but it is an electrostatic signal, moving essentially at the speed of light. Second, there is no evidence for timing being used by these electroreceptors to measure speed of objects. The idea that appears to be missing here (and possibly the idea that the authors may be going for?) is that high frequency tuning is matched to the power spectrum of the EOD, and that the Knollenorgan sensory pathway involves specializations (electrical synapses, large diameter fibers, synapses on cell bodies etc.) so that there is very little jitter in the temporal coding of the EOD. This lack of jitter permits the EOD to be recognized by the timing cues. But that is different from the "microsecond timing difference" of sound localization, which seems to be referenced here.

We agree and we have removed this from the text.

8. Line 31 and line 67 "Raman and Carlson." The paper is by Raman and Hopkins, not Raman

and Carlson.

Text revised as suggested.

9. Lines 33-34. "The authors hypothesize that the mechanisms are linear bandpass filtering coupled with non-linear thresholding..." This statement is incorrect.

a. Contrary to the statement made by the authors, Figure 1 of the paper shows that a simple bandpass filter is *not* sufficient to describe the tuning curves. It shows that there is instead some process that mimics an electrical resonance. This should be corrected.

b. The circuit in the perspective's Figure 1B is the diagram proposed (by Carlson) in Lyons-Warren et al. (2012), for *B. brachyistius*, a species not studied in the Raman-Hopkins paper. This circuit is *not* able to describe the results in the present paper. Therefore, the statement in lines 33-34 should be modified or corrected, and the circuit diagram probably should be changed or removed.

a. We have removed this sentence and in the second paragraph we state: "These tuning curves matched very well with the spectrum of EODs these fish emit, a strong indication that this electrical tuning has evolved to mediate conspecific communication. Moreover, a combination nonlinear thresholding and rectification produces well-timed spikes in Knollenorgan electroreceptors that vary in waveform across different species of weakly electric fish. These electrically evoked spikes in Knollenorgans are reliable and precise on a time scale of less than 0.1 ms and can thus transmit signals above 10 kHz."

b. We changed the R_m in the Figure 1B to $R_m(V)$ to indicate that it is voltage dependent.

10. Lines 35-37. "they propose a subthreshold electrical resonance in Knollenorgan electroreceptors that stems from Ca^{2+} currents closely coupled with Ca^{2+} -dependent K^+ currents." This statement is inaccurate. No model of specific currents is proposed. It should be made clear in the text that the present paper makes no claims about mechanism because at present no mechanistic data are available. The paper (by intention) presents the phenomenology that directs the hunt for mechanism. The hypotheses tested in the paper were (1) that the high frequencies of the EODs of these three species would be predictive of correspondingly high-frequency tuned Knollenorgans (supported), and (2) that the transduction process might be approximated as a linear filter (supported with the revision of including rectification and the non-linear spike generating mechanism). These hypotheses should be distinguished from the subsequent questions that became possible to articulate once the paper's results were in hand, namely, how (mechanistically) can electroreceptors achieve tuning up to such high frequencies? The data presented in the paper offer a constraint, namely, whatever the mechanism actually is, its output must produce something that mimics a bandpass filter scaled by an electrical resonance. The relevant section of the Discussion begins with the statement "The puzzle that persists is mechanistic," followed by a literature review of pertinent information from other systems. Ca and $K(Ca)$ currents are discussed in that context, but no claims or proposals are made in the absence of data.

We revised the text to avoid saying "they propose...". Instead, we are simply explaining how we think a potential mechanism could produce the data.

11. Lines 49-51: "Figure 1B shows a putative intracellular recording from a single electroreceptor." This is a far overstated claim. This is not a putative intracellular recording, but is the output of a model proposed for a different species, and the only data available (i.e., the Raman-Hopkins paper) indicate a very different mechanism. Along the lines of the previous point, an acknowledgment of species diversity is crucial in the context of offering perspective on

a paper that is about species diversity.

We extensively revised the text to show that how we think the electrical tuning described in the study may occur in a simple model with electrical resonance and spike thresholds. We now mention and emphasize the species diversity issue.

12. Figure 1A: it is not clear where that schematic came from. A reference is needed. It may be worth noting that the sensory cells in the EMs that from Szabo have microvilli all around, not just on the apical end. (In this regard they are unlike mechanosensitive hair cells).

We have checked the morphology and modified the diagram as suggested.

13. Minor. Line 9-10. "murky and dark rivers of Africa." The authors might consider "murky and dim rivers."

Text revised as suggested.

Dear Dr Li,

Re: JP-P-2025-289354R1 "**Electroreceptors with a different kind of buzz**" by Henrique von Gersdorff and Geng-Lin Li

Thank you for submitting your manuscript to The Journal of Physiology. It has been assessed by a Reviewing Editor and by 1 expert referee.

There are still major revisions to be made. We would like to give you the opportunity of responding to the reviewer comments and submitting a further revised version for consideration.

If you decide you do not wish to do this, please let us know and we can withdraw the submission for you.

The review comments are copied at the end of this email.

Please address all the points raised and incorporate all requested revisions or explain in your Response to Referees why a change has not been made. We hope you will find the comments helpful and that you will be able to return your revised manuscript within 2 weeks. If you require longer than this, please contact journal staff: jp@physoc.org. Please note that this letter does not constitute a guarantee for acceptance of your revised manuscript.

REVISION CHECKLIST:

IMPORTANT POINTS TO NOTE WHEN REVISING YOUR MANUSCRIPT:

LANGUAGE EDITING AND SUPPORT FOR PUBLICATION: If you would like help with English language editing, or other article preparation support, Wiley Editing Services offers expert help, including English Language Editing, as well as translation, manuscript formatting, and figure formatting at www.wileyauthors.com/eoo/preparation. You can also find resources for Preparing Your Article for general guidance about writing and preparing your manuscript at www.wileyauthors.com/eoo/prepresources.

We look forward to receiving your revised submission.

Yours sincerely,

Katalin Toth
Senior Editor
The Journal of Physiology

EDITOR COMMENTS

Reviewing Editor:

There are still significant issues that authors need to respond to (see reviewer comments below). Please respond to the critiques and address the concerns with a significant rewrite.

REFEREE COMMENTS

Referee #1:

The Comment is improved in terms of removing several mistakes, but we remain somewhat perplexed in reading it. The authors still seem to be intent on proposing theories about how resonance might come about, and less on discussing the content of the paper which is focused on the discovery that receptor tuning varies by species and is correlated with the power spectrum of the EOD. Nevertheless, we recognize that that is the choice of the authors and the decision of the editors.

1. General comment: The focus on a simple single compartmental model seems to both of us (Raman and Hopkins) to shortchange the Discussion section of the paper itself, where additional factors beyond extraordinary biophysical capabilities are considered. It is particularly surprising that the present authors focus on the capacitance C_m of the entire Knollenorgan receptor cell rather than its obvious subcomponents which are the microvilli that completely cover the apical surface of the cell and could well be a tiny resonant structure separate from the main cell body. Why couldn't the microvilli be considered an important secondary resonant structure? (Think of a violin with a large chamber that resonates with low frequencies and a smaller chamber for high frequencies.) In any case, it is somewhat odd to read through speculations on how a single mechanism might work, especially when it doesn't relate to the original publication. (Perhaps these ideas would be better in a separate theoretical paper on that topic rather than a comment on this paper?)

2. Factual: The diagram presented is not grounded in what is physically known, and seems to take the analogy between mechanosensitive hair cells and Knollenorgan electroreceptors too far, ignoring physical constraints that can guide speculations. One point of interest is that in the drawing they do not indicate how the current through the pore of the receptor is directed to cross the cell membrane of the receptor instead of flowing freely around the cell. The thing missing here is the attachment of the base of the receptor cell to other cell types that might form a current barrier that directs the current through the apical face, and then through the basal face where the synapse occurs. This cell type at the basal face of the receptor is called a basal accessory cell or cells, described in Szabo's 1974 Handbook of Sensory Physiology chapter on the anatomy of the Knollenorgan as the basal accessory cells that form a hillock that attach to the base of the Knollenorgan. Not much is known about this cell type, nor about the resistance of this cell layer, but it is hard to imagine how the receptor cell is activated by current unless it forms a resistive barrier that forces the current through the membrane of the receptor cell causing a depolarization that activates voltage gated calcium channels in the membrane of the cell.

3. Factual: Another specific point is that the figure and the circuit diagram they present are somewhat confusing/misleading. The structures on the surface of the receptor cell are labeled as "cilia", but they should be called microvilli since there are no cilia within the microvilli.

4. Factual: "Moreover, a combination nonlinear thresholding and rectification produces well-timed spikes in Knollenorgan electroreceptors that vary in waveform across different species of weakly electric fish." There must be a cut-and-paste error here, since part of the (presumed) information is lost. (The nonlinear thresholding and rectification don't produce spikes that vary in waveform.) Perhaps what is meant could be written simply as "Moreover, arbitrary electrical noise stimuli applied to Knollenorgan electroreceptors evoked precisely timed spike patterns, which were well replicated by a model of linear filtering, rectification, and spike thresholding."

5. Factual: "Remarkably, Knollenorgans fire spikes whose waveforms are diverse among different species, as shown by

Raman and Hopkins (2025)." Spike waveforms were not recorded and not shown in our study. Perhaps this can be amended to "Remarkably, Knollenorgans fire spikes whose waveforms are diverse among different species, as discussed by Raman and Hopkins (2025)."

6. Minor typos: "resonance frequency" should probably be "resonant frequency" and "Pangea" probably should be "Pangaea."

7. The "murky and dim" line still has a little bit of a dramatic quality that reads awkwardly in the present era. Style is of course the authors' prerogative, but it might be more straightforward just to start with a more scientific line like, "Electric fish have evolved electroreceptors and electric organs that permit them to communicate, navigate, and find prey even as nocturnal animals inhabiting muddy rivers."

END OF COMMENTS

Dear Dr. Toth,

Thank you very much for forwarding us the review comments (in blue), and please find our point-by-point response below (in black). All changes to the text of our Perspective piece are marked in red (marked copy). We hope you and the reviewers find the manuscript in the current version acceptable for publishing.

EDITOR COMMENTS

Reviewing Editor:

There are still significant issues that authors need to respond to (see reviewer comments below). Please respond to the critiques and address the concerns with a significant rewrite. See our point-by-point response below. We have changed the text significantly. We believe our Perspective is well written now and it clearly highlights the main findings of the Raman and Hopkins paper. Moreover, it includes many novel ideas and hypotheses that go beyond the paper, making it hopefully a more stimulating read that will inspire the readers of the J. of Physiology to read the paper of Raman and Hopkins.

REFEREE COMMENTS

Referee #1:

The Comment is improved in terms of removing several mistakes, but we remain somewhat perplexed in reading it. The authors still seem to be intent on proposing theories about how resonance might come about, and less on discussing the content of the paper which is focused on the discovery that receptor tuning varies by species and is correlated with the power spectrum of the EOD. Nevertheless, we recognize that that is the choice of the authors and the decision of the editors.

The findings in Raman and Hopkins (2025) are very interesting and significant. Besides giving a short introduction to electric fish electroreceptors for the non-specialist (see our Figure 1), our Perspective highlights many of the main findings of Raman and Hopkins (2025). With this Perspective, we attempt to bring these findings to attention of a much broader audience, including cellular physiologists and biophysicists, who are probably not familiar with the topic. We feel a good Perspective piece should provide also new hypotheses and speculations about mechanisms that are not mentioned in the main paper. This piques interest in the main paper and motivates further investigations in the future, as researchers become aware of many fascinating and unresolved issues still left open by the main paper. We feel this makes for a much more interesting Perspective for the general reader.

1. General comment: The focus on a simple single compartmental model seems to both of us (Raman and Hopkins) to shortchange the Discussion section of the paper itself, where additional factors beyond extraordinary biophysical capabilities are considered. It is particularly surprising that the present authors focus on the capacitance C_m of the entire Knollenorgan receptor cell rather than its obvious subcomponents which are the microvilli that completely cover the apical surface of the cell and could well be a tiny resonant structure separate from the main cell body. Why couldn't the microvilli be considered an important secondary resonant structure? (Think of a violin with a large chamber that resonates with low frequencies and a smaller chamber for high frequencies.) In any case, it is somewhat odd to read through speculations on how a single mechanism might work, especially when it doesn't relate to the original publication. (Perhaps these ideas would be better in a separate theoretical paper on that topic rather than a comment on this paper?)

We are severely limited by the number of words in the Perspective so we cannot include these new ideas in the writing. Unless the microvilli act somehow independently from the soma, the fast and small signals in the microvilli are doomed to be slowed and overwhelmed by the rather

large overall soma capacitance and presumably low membrane resistance. It is not clear how the microvilli could act as "... resonant structure separate from the main cell body". Therefore, our simple cellular model with abundant ion channels to drive the voltage changes seems more plausible as a first attempt to understand the mechanisms underlying resonance. These new ideas are very interesting and deserve a future modeling paper, but they are beyond the scope of a word-limited Perspective.

2. Factual: The diagram presented is not grounded in what is physically known, and seems to take the analogy between mechanosensitive hair cells and Knollenorgan electroreceptors too far, ignoring physical constraints that can guide speculations. One point of interest is that in the drawing they do not indicate how the current through the pore of the receptor is directed to cross the cell membrane of the receptor instead of flowing freely around the cell. The thing missing here is the attachment of the base of the receptor cell to other cell types that might form a current barrier that directs the current through the apical face, and then through the basal face where the synapse occurs. This cell type at the basal face of the receptor is called a basal accessory cell or cells, described in Szabo's 1974 Handbook of Sensory Physiology chapter on the anatomy of the Knollenorgan as the basal accessory cells that form a hillock that attach to the base of the Knollenorgan. Not much is known about this cell type, nor about the resistance of this cell layer, but it is hard to imagine how the receptor cell is activated by current unless it forms a resistive barrier that forces the current through the membrane of the receptor cell causing a depolarization that activates voltage gated calcium channels in the membrane of the cell.

The point of the diagram is to show that two capacitors and two resistors in a simple circuit can produce an overall V-shaped tuning curve (see figure caption). This is not an obvious idea to most people, and this diagram comes from several previous studies of the Knollenorgan (see Lyons-Warren et al., 2012). A clear and simple circuit diagram should highlight just the main circuit elements, and it implies that current must flow through the electroreceptor to depolarize the cell and open its Ca channels. It is simplified to convey the main ideas.

3. Factual: Another specific point is that the figure and the circuit diagram they present are somewhat confusing/misleading. The structures on the surface of the receptor cell are labeled as "cilia", but they should be called microvilli since there are no cilia within the microvilli. Corrected as suggested.

4. Factual: "Moreover, a combination nonlinear thresholding and rectification produces well-timed spikes in Knollenorgan electroreceptors that vary in waveform across different species of weakly electric fish." There must be a cut-and-paste error here, since part of the (presumed) information is lost. (The nonlinear thresholding and rectification don't produce spikes that vary in waveform.) Perhaps what is meant could be written simply as "Moreover, arbitrary electrical noise stimuli applied to Knollenorgan electroreceptors evoked precisely timed spike patterns, which were well replicated by a model of linear filtering, rectification, and spike thresholding." This is a very helpful suggestion. We edited the sentence as suggested.

5. Factual: "Remarkably, Knollenorgans fire spikes whose waveforms are diverse among different species, as shown by Raman and Hopkins (2025)." Spike waveforms were not recorded and not shown in our study. Perhaps this can be amended to "Remarkably, Knollenorgans fire spikes whose waveforms are diverse among different species, as discussed by Raman and Hopkins (2025)." We have changed "shown" to "discussed".

6. Minor typos: "resonance frequency" should probably be "resonant frequency" and "Pangea"

probably should be "Pangaea."

We have changed "resonance frequency" to "resonant frequency" where appropriate, and regarding "Pangea" vs. "Pangaea", both spellings are acceptable and used in the literature.

7. The "murky and dim" line still has a little bit of a dramatic quality that reads awkwardly in the present era. Style is of course the authors' prerogative, but it might be more straightforward just to start with a more scientific line like, "Electric fish have evolved electroreceptors and electric organs that permit them to communicate, navigate, and find prey even as nocturnal animals inhabiting muddy rivers."

We thank the reviewers for this helpful suggestion. We have changed the first two sentences to read: "Electric fish in Africa and South America have evolved electroreceptors and electric organs that permit them to detect and produce electric fields. They use these electric fields to communicate, navigate, and find prey even as nocturnal animals inhabiting muddy rivers where visual cues are limited."

Dear Dr Li,

Re: JP-P-2025-289354R2 "**Electroreceptors with a different kind of buzz**" by Henrique von Gersdorff and Geng-Lin Li

We are pleased to tell you that your paper has been accepted for publication in The Journal of Physiology.

Please note that Perspective articles are not typically covered by institutional open access agreements with our publisher, Wiley. Wiley do not offer article processing charge (APC) discounts for smaller article types in hybrid subscription journals, meaning that if you wish for your Perspective to be published Open Access, you will have to pay the full APC. As such, we recommend authors publish Perspectives 'behind the paywall', where they will become freely accessible after a 12-month embargo (i.e. please select the NON open access option via Wiley Author services during proofing).

Should you wish to pay for Open Access, you will be able to place an order by logging into Wiley Author services.

Yours sincerely,

Katalin Toth
Senior Editor
The Journal of Physiology

IMPORTANT POINTS TO NOTE FOLLOWING ACCEPTANCE OF YOUR PAPER:

- You can help your research get the attention it deserves! Check out Wiley's free Promotion Guide for best-practice recommendations for promoting your work at: www.wileyauthors.com/eoo/guide. You can learn more about Wiley Editing Services which offers professional video, design, and writing services to create shareable video abstracts, infographics, conference posters, lay summaries, and research news stories for your research at: www.wileyauthors.com/eoo/promotion.

- If you would like to receive our 'Research Roundup', a monthly newsletter highlighting the cutting-edge research published in The Physiological Society's family of journals (The Journal of Physiology, Experimental Physiology, Physiological Reports, The Journal of Nutritional Physiology and The Journal of Precision Medicine: Health and Disease), please click this link, fill in your name and email address and select 'Research Roundup': <https://www.physoc.org/journals-and-media/membernews>

EDITOR COMMENTS

Reviewing Editor:

The authors have done a satisfactorily job of responding to the reviewer comments and critiques.

REFEREE COMMENTS

Referee #1:

Thank you for the opportunity to re-review this Perspective. We have no further comments.